# Determination of the lipid composition of the GPI anchor

**Auxiliadora Aguilera-Romero**[1☯]*, **Susana Sabido-Bozo**[1☯], **Sergio Lopez**[1],
**Alejandro Cortes-Gomez**[1], **Sofia Rodriguez-Gallardo**[1], **Ana Maria Perez-Linero**[1],
**Isabelle Riezman**[2], **Howard Riezman**[2], **Manuel Muñiz**[1]*

**1** Departamento de Biología Celular, Facultad de Biología, Universidad de Sevilla e Instituto de Biomedicina de Sevilla (IBiS), Hospital Universitario Virgen del Rocío/CSIC/Universidad de Sevilla, Seville, Spain, **2** Department of Biochemistry, NCCR Chemical Biology, University of Geneva, Geneva, Switzerland

☯ These authors contributed equally to this work.

* auxi@us.es (AAR); mmuniz@us.es (MM)

**Data Availability Statement:** All relevant data are within the manuscript and its Supporting Information files.

## Abstract

In eukaryotic cells, a subset of cell surface proteins is attached by the glycolipid glycosylphosphatidylinositol (GPI) to the external leaflet of the plasma membrane where they play important roles as enzymes, receptors, or adhesion molecules. Here we present a protocol for purification and mass spectrometry analysis of the lipid moiety of individual GPI-anchored proteins (GPI-APs) in yeast. The method involves the expression of a specific GPI-AP tagged with GFP, solubilization, immunoprecipitation, separation by electrophoresis, blotting onto PVDF, release and extraction of the GPI-lipid moiety and analysis by mass spectrometry. By using this protocol, we could determine the precise GPI-lipid structure of the GPI-AP Gas1-GFP in a modified yeast strain. This protocol can be used to identify the lipid composition of the GPI anchor of distinct GPI-APs from yeast to mammals and can be adapted to determine other types of protein lipidation.

## Introduction

Lipidation is an essential post-translational modification by which proteins are covalently modified with specific lipids that regulate protein localization, function, and stability. GPI anchoring is a special type of lipidation present in all eukaryotes that occurs at the endoplasmic reticulum (ER) and targets GPI-anchored proteins (GPI-APs) to the cell surface where they play a wide variety of essential physiological roles [1]. Immediately after GPI attachment and during GPI-AP secretory transport to the cell surface, the lipid moiety of the GPI anchor (GPI-lipid) undergoes structural remodeling, which is important for GPI-AP function and trafficking [2]. The initial structure and subsequent remodeling process of the precursor GPI-lipid moiety vary among proteins and species. In the yeast *Saccharomyces cerevisiae*, for instance, two different types of lipid moieties, diacylglycerol or ceramide, can be present in the GPI anchors on mature proteins [3].

We have used mass spectrometry to directly elucidate the structure of the GPI-lipid moiety of the specific GPI-AP, Gas1 [4]. Mass spectrometry-based methods have been extensively

**Funding:** This research was funded by the FEDER/ Ministerio de Ciencia, Innovación y Universidades —Agencia Estatal de Investigación/BFU2017- 89700-P to Manuel Muñiz, "VI Own Research Plan" of the University of Seville VIPPIT-2020-I.5 to Manuel Muñiz, by the Incentivo al Grupo de Investigación BIO-271 (2019/BIO-271) and the NCCR Chemical Biology and the Swiss National Science Foundation (51NF40-185898 and 310030_184949) to Howard Riezman. 'V Own Research Plan' of the University of Seville (VPPI- US) contract (cofounded by the European Social Fund) to Sergio Lopez, University of Seville fellowships to Sofia Rodriguez-Gallardo and Ana Maria Perez-Linero, Ministry of Education, Culture and Sport (MECD) fellowship to Susana Sabido- Bozo and contract from the University of Seville by the Youth Employment Initiative to Alejandro Cortes-Gomez. The funders had and will not have a role in study design, data collection and analysis, decision to publish, or preparation of the manuscript.

**Competing interests:** The authors have declared that no competing interests exist.

used for detailed analysis of posttranslational modifications. This methodology presents important advantages for GPI-lipid moiety analysis, such as higher precision and ease of use than classical techniques based on the incorporation of radioactive lipid precursors and subsequent detection of the proteins by fluorography [5]. Since identification by mass spectrometry of a single yeast GPI-AP, such as Gas1, requires prior purification of the protein, we need either a specific antibody or to use Gas1 tagged with GFP for immunopurification [6]. The latter makes the protocol applicable to a wider range of protein substrates. Yeast cells expressing Gas1-GFP are broken by glass beads and differentially centrifuged to generate a membrane fraction. Gas1-GFP is solubilized from the membrane fraction using the detergent digitonin and then affinity purified by the GFP-trap system. The immunopurified proteins are separated by SDS-polyacrylamide gel electrophoresis (SDS-PAGE), which contributes to completely remove any remnant of membrane lipids from the protein sample. Next, electrophoresed Gas1-GFP is transferred and immobilized on activated polyvinylidenedifluoride (PVDF) membrane by Western blotting. The PVDF membrane is treated with nitrous acid to deaminate the glucosamine residue of the GPI-glycan, a residue which occurs only rarely in mammalian glycans, but is part of the core structure, common to all GPIs. This chemical reaction is specific for the amino sugar and cleaves the protein and glycan attached to the GPI-glycan from the inositol-containing GPI-lipid [7], which remains attached to the PVDF membrane due to its hydrophobic nature. Finally, the GPI-lipid can be recovered by extraction with the negative mode solvent from the PVDF membrane for analysis by ESI-MS/MS mass spectrometry. For lipid species identification we use a multiple reaction monitoring (MRM) approach [8, 9].

The MRM is a targeted method used in tandem mass spectrometry. For each lipid molecular species, a parent ion and a product ion are defined. The parent ion is selected and fragmented to create a product ion that is finally detected. This is a highly selective and sensitive method that allows a rapid lipid profiling of the GPI-lipid. The disadvantage of the MRM is that requires a user-defined list of lipid species. Therefore, to decipher an uncharacterized lipid moiety a nontargeted profiling approach would be more suitable.

## Materials and methods

"The protocol described in this peer-reviewed article is published on protocols.io **https://dx. doi.org/10.17504/protocols.io.bvban2ie** and is included for printing as S1 File with this article.

## Expected results

The protocol described here for analyzing the lipid composition of the GPI anchor was employed to address the role of the chain length of membrane ceramide in the sorting of GPI-APs upon exit from the endoplasmic reticulum (ER) in yeast [4]. In wild type cells, very long (C26) chain ceramide is present in the ER membrane and in the lipid moiety of the GPI anchor of the GPI-AP Gas1 [10]. We showed that newly synthesized Gas1-GFP is segregated from transmembrane secretory proteins and sorted into selective ER exit sites (ERES) during ER export. To tackle the relevance of membrane ceramide for ER sorting, we engineered a modified yeast strain (GhLag1) that produces cellular membranes with shorter ceramides (C16-C18) than in the wild-type strain (C26) [4, 11]. Mass spectrometry analysis revealed that although C18 and C16 ceramides are by far the major ceramides detected in GhLag1 membranes, the GPI anchor of Gas1-GFP expressed in GhLag1 strain contains C26 ceramide, the same lipid, as in wild-type [4]. Therefore, because in the GhLag1 strain the acyl chain length of membrane ceramides, but not the GPI ceramide is decreased, we could use this strain to

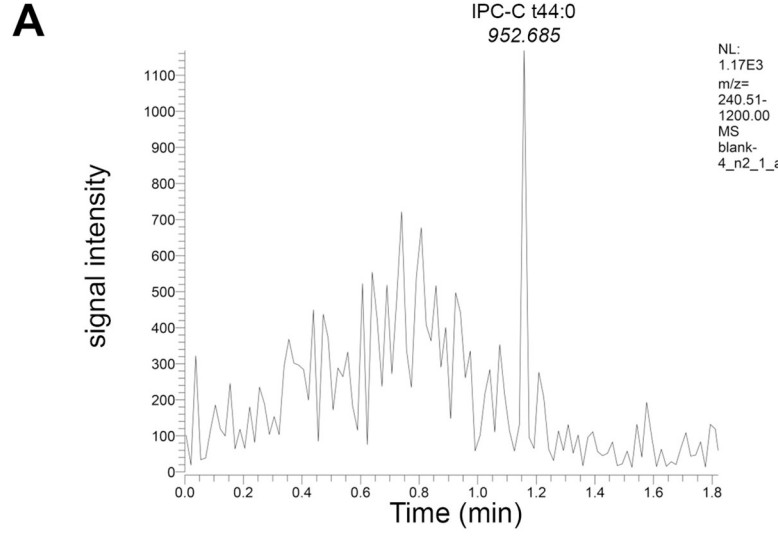

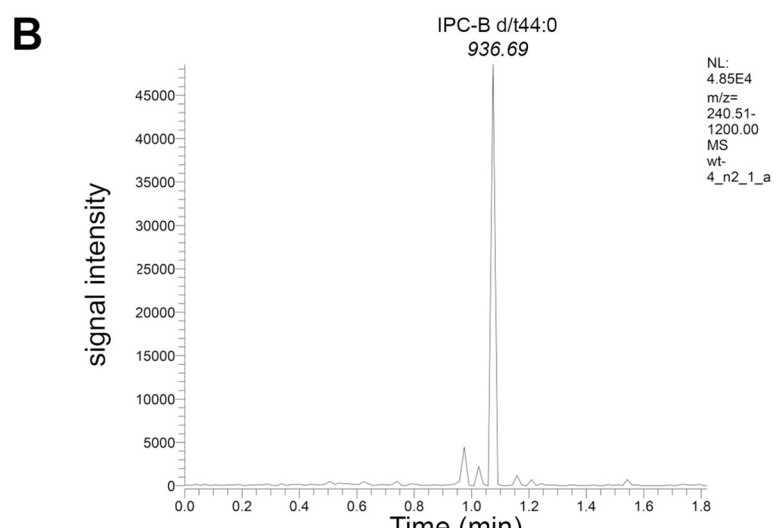

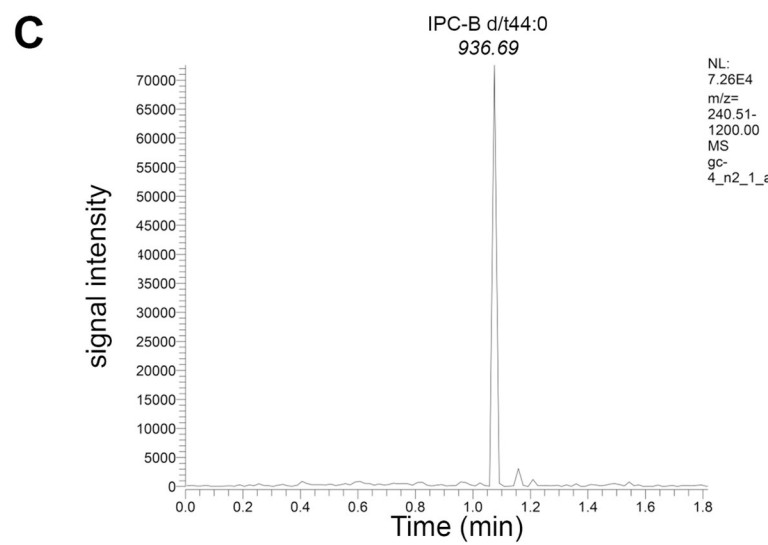

**Fig 1. Mass spectrometry analysis of GPI lipid species of Gas1-GFP in wild-type and GhLag1 cells.** Isolated GPI lipids of Gas1-GFP from wild-type (B) and GhLag1 (C) cells and the blank sample (A) were detected by electrospray ionization and tandem mass spectrometry. The figure shows a representative cycle of transition measurements. The signal intensity of each lipid species was detected by multiple reaction monitoring. The GPI lipid from wild-type cells and Ghlag1 cells was identified as IPC with phytosphingosine and saturated C26 fatty acid (IPC-B d/t44:0) Time indicates the reading moment of each transition.

specifically examine the role of the acyl chain length of membrane ceramides in ER sorting. We observed that Gas1-GFP expressed in GhLag1 failed to be sorted into selective ERES and, instead, was rerouted to exit the ER with transmembrane secretory proteins via common ERES. Based on these results, we concluded that ceramide acyl chain length in the ER membrane is an essential determinant for ER protein clustering and sorting [4].

We determined that in GhLag1 strain the GPI anchor of Gas1-GFP contains C26 ceramide using the method described here. For this purpose, Gas1-GFP was expressed under control of its own promoter in wild-type and GhLag1 cells and was then immunopurified from solubilized membranes with the GFP-trap system, applied to SDS-PAGE, transferred to a hydrophobic PVDF membrane and stained with amido black. The respective bands were cut and treated with nitrous acid to separate the inositolphosphoceramide (IPC) from the Gas1-GFP protein and the GPI-glycan. The separated IPC, that remains attached to the PVDF membrane due to its hydrophobicity, was extracted using the negative solvent for analysis by negative ion ES-MS/MS mass spectrometry. The sample was infused on a TSQ Vantage using a Triversa Nanomate. To provide a fast and accurate profile of the Gas1p lipid anchor we use an MRM approach applying a predefined list of known inositolphosphoceramide lipids of the yeast *Saccharomyces cerevisiae* (S1 Table). To obtain the published results, 5 biological replicates of blank, wild-type and Ghlag1 strains were done and the MRM list was read 3 times for each of them (technical replicates). As seen in Fig 1, the GPI-lipid of Gas1-GFP from wild-type and Ghlag1 strains specifically contains the lipid species IPC-B d/t44:0, an IPC with phytosphingosine (4-hydroxysphinganine) and a saturated C26 fatty acid. Therefore, our protocol provides qualitative data that shows that the GPI-lipid of Gas1 in the strain GhLag1 has a C26 ceramide. This method was developed for ceramide-type GPI-APs however, this could also be applied for a diacylglycerol-type phosphatidylinositol GPI-APs of yeast using the MRM list provided (S2 Table).

## Supporting information

**S1 File. Step-by-step protocol, also available on protocols.io.**
(PDF)

**S1 Table. *m/z* of precursor/product ion for multiple-reaction monitoring of sphingolipid species of the yeast *Saccharomyces cerevisiae* used in this study.**
(PDF)

**S2 Table. *m/z* of precursor/product ion for multiple-reaction monitoring of phosphatidylinositol lipid species from the yeast *Saccharomyces cerevisiae*.**
(PDF)

## Author Contributions

**Conceptualization:** Manuel Muñiz.

**Investigation:** Auxiliadora Aguilera-Romero, Susana Sabido-Bozo, Sergio Lopez, Alejandro Cortes-Gomez, Sofia Rodriguez-Gallardo, Ana Maria Perez-Linero, Isabelle Riezman, Howard Riezman.

**Methodology:** Auxiliadora Aguilera-Romero, Susana Sabido-Bozo, Sergio Lopez, Alejandro Cortes-Gomez, Sofia Rodriguez-Gallardo, Ana Maria Perez-Linero, Isabelle Riezman, Howard Riezman.

**Supervision:** Auxiliadora Aguilera-Romero, Howard Riezman, Manuel Muñiz.

**Writing – original draft:** Auxiliadora Aguilera-Romero, Manuel Muñiz.

**Writing – review & editing:** Susana Sabido-Bozo, Isabelle Riezman, Howard Riezman.

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
