## [Decision Letter · Decision Letter 0]

22 Jun 2021

PONE-D-21-17892

Determination of the lipid composition of the GPI anchor

PLOS ONE

Dear Dr. Muñiz,

Thank you for submitting your manuscript to PLOS ONE. After careful consideration, we feel that it has merit but does not fully meet PLOS ONE’s publication criteria as it currently stands. Therefore, we invite you to submit a revised version of the manuscript that addresses the points raised during the review process.

As you can see, the suggestions are easy to address with text modifications and they will improve the manuscript.

We look forward to receiving your revised manuscript.

Kind regards,

Michael Polymenis, Ph.D.

Academic Editor

PLOS ONE

Journal Requirements:

2. Please include your tables as part of your main manuscript and remove the individual files. Please note that supplementary tables should remain uploaded as separate "supporting information" files.

"This research was funded by the FEDER/Ministerio de Ciencia, Innovación y

Universidades—Agencia Estatal de Investigación/BFU2017-89700-P to Manuel Muñiz, "VI

Own Research Plan" of the University of Seville VIPPIT-2020-I.5 to Manuel Muñiz, by the

Incentivo al Grupo de Investigación BIO-271 (2019/BIO-271), and the NCCR Chemical

Biology and the Swiss National Science Foundation (51NF40-185898 and

310030_184949) to Howard Riezman. ‘V Own Research Plan’ of the University of Seville

(VPPI-US) contract (cofounded by the European Social Fund) to Sergio Lopez, University

of Seville fellowships to Sofia Rodriguez-Gallardo and Ana Maria Perez-Linero, Ministry of

Education, Culture and Sport (MECD) fellowship to Susana Sabido-Bozo and contract from

the University of Seville by the Youth Employment Initiative to Alejandro Cortes-Gomez

Competing interests:"

Reviewers' comments:

Reviewer's Responses to Questions

**Comments to the Author**

1. Does the manuscript report a protocol which is of utility to the research community and adds value to the published literature?

Reviewer #1: Yes

Reviewer #2: Yes

Reviewer #3: Yes

2. Has the protocol been described in sufficient detail?

Descriptions of methods and reagents contained in the step-by-step protocol should be reported in sufficient detail for another researcher to reproduce all experiments and analyses. The protocol should describe the appropriate controls, sample sizes and replication needed to ensure that the data are robust and reproducible.

Reviewer #1: Yes

Reviewer #2: Yes

Reviewer #3: Yes

3. Does the protocol describe a validated method?

Reviewer #1: Yes

Reviewer #2: Yes

Reviewer #3: Yes

4. If the manuscript contains new data, have the authors made this data fully available?

Reviewer #1: Yes

Reviewer #2: Yes

Reviewer #3: Yes

**5. Is the article presented in an intelligible fashion and written in standard English?**

Reviewer #1: Yes

Reviewer #2: Yes

Reviewer #3: Yes

6. Review Comments to the Author

Reviewer #1: The protocol article written by Aguilera-Romero et al described the lipid analysis of GPI-anchored proteins. The authors showed an analytical method of inositol phosphoceramide structures on GFP-tagged Gas1 proteins expressed in Saccharomyces cerevisiae. It would be useful for readers who are interested in lipid species of GPI-anchors and their changes by mutations in genes. The reviewer recommends the publication after the appropriate revision.

1) In the abstract, the authors wrote “an improved protocol”. Please describe more clearly what is improved compared to the previous methods (Fontaine et al. (2003) Glycobiology 13(3):169-77; Yoko-o et al. (2013) Mol. Microbiol. 88(1):140-55). Compared to the old methods, purification of PI using silica column was omitted. Is it because of changing the detection method using MRM in the analysis by ESI-MS/MS?

2) The authors listed the ions for MRM of lipids in Table 1. In the list, not only IPC species, but also MIPC and M(IP)2C species are listed. MIPC and M(IP)2C should be removed from the list, since these lipid species are not observed in the GPI lipids. Instead, is it possible to add diacylglycerol-type PI species?

Reviewer #2: This protocol, presented here at the detail of a laboratory protocol, provides a highly standard series of methods to identify the lipid attached to an immunoaffinity tagged GPI-anchored protein. The force of these methods to identify lipid attachments to GPI-proteins is shown fully in their recent publication in Science Advances 2020;6(49). Developed in yeast, such is the importance of identifying lipid modification in all species that I am confident this protocol will be used widely in the future.

Reviewer #3: The authors provide an excellent job of summarizing the significance and relevance of improving the protocol for detection of GPI-AP in yeast. This includes a detailed explanation of the roles of GPI and the distinctions between that of yeast and mammalian. Additionally, a comprehensive description of the rational for use of mass spectrometry-based methods versus the use of traditional methods. The authors' observations regarding the significance of using the ceramide acyl chain length for the ER protein. There are a few clarifications that I am looking for:

1) How many samples were used for each the wild type and GhLag1, and just one blank were used? Were any biological or technical replicated used?

2) Which instrument/columns were used for MS/MS?

Thank you.

7. PLOS authors have the option to publish the peer review history of their article (what does this mean?). If published, this will include your full peer review and any attached files.

Reviewer #1: No

Reviewer #2: **Yes: **Roger J Morris

Reviewer #3: No

---

## [Author Response · Author response to Decision Letter 0]

21 Jul 2021

Point by Point response

Reviewer #1:

The protocol article written by Aguilera-Romero et al described the lipid analysis of GPI-anchored proteins. The authors showed an analytical method of inositol phosphoceramide structures on GFP-tagged Gas1 proteins expressed in Saccharomyces cerevisiae. It would be useful for readers who are interested in lipid species of GPI-anchors and their changes by mutations in genes. The reviewer recommends the publication after the appropriate revision.

We thank the reviewer for the very positive and helpful comments on our manuscript, and we hope to have answered all concerns raised.

1) In the abstract, the authors wrote “an improved protocol”. Please describe more clearly what is improved compared to the previous methods (Fontaine et al. (2003) Glycobiology 13(3):169-77; Yoko-o et al. (2013) Mol. Microbiol. 88(1):140-55). Compared to the old methods, purification of PI using silica column was omitted. Is it because of changing the detection method using MRM in the analysis by ESI-MS/MS?

Response: As pointed out by the reviewer, the adjective “improved” is not appropriate because our protocol is a compilation of methods used in Fontaine et al. (2003) Glycobiology 13(3):169-77, Yoko-o et al. (2013) Mol. Microbiol. 88(1):140-55 and Mehlert et al. (2009) Glycoconj J. 26(8):915-921. Therefore, we remove this adjective from the abstract. Following the method used by Mehlert et al. (2009) we decided to directly infuse the sample on the MS/MS without a previous purification by silica column. Thanks to the use of a targeted approach, MRM, we could clearly identify the lipid specie of the GPI anchor despite of background noise. This simplifies the method allowing more routine studies. However, for a more explorative approach we consider the protocol described in Yoko-o et al. (2013) to be more suitable. The text is now corrected as suggested by the reviewer.

2) The authors listed the ions for MRM of lipids in Table 1. In the list, not only IPC species, but also MIPC and M(IP)2C species are listed. MIPC and M(IP)2C should be removed from the list since these lipid species are not observed in the GPI lipids. Instead, is it possible to add diacylglycerol-type PI species?

Response: In Rodriguez-Gallardo et al. (2020) (Sci Adv 6(50):eaba8237) we studied the role of very long (C26) chain membrane ceramides in protein sorting from the ER. For this purpose, we used a modified yeast strain GhLag1, that produces cellular membranes with only long (C18) rather than very long (C26) chain ceramides (Epstein et al. (2012) Mol Microbiol 84, 1018). Gas1, used in this work as a reporter, is a GPI-AP that has a C26 ceramide lipid in its GPI-anchor (Yoko-o et al. (2013) Mol. Microbiol. 88(1):140-55). Since the strain GhLag1 mainly produces C18 ceramides we wanted to determine whether Gas1 incorporates a C18 or a C26 ceramide in its GPI anchor. To achieve this, we developed the present method and applied a MRM list used routinely in our lab to define the sphingolipid composition of yeast strains, which contains IPC, MIPC and MIP2C lipid species. We agree with the reviewer comment that MIPC and MIP2C species are not required for our purpose, but we want to show the exact method used in the published paper that corresponds with the figure provided. Additionally, we think that the suggestion provided by the reviewer could be of interest to determine the lipid of the GPI-anchor therefore we provide an additional S2 table that contains a diacylglycerol-type PI MRM list.

Reviewer #2: 

This protocol, presented here at the detail of a laboratory protocol, provides a highly standard series of methods to identify the lipid attached to an immunoaffinity tagged GPI-anchored protein. The force of these methods to identify lipid attachments to GPI-proteins is shown fully in their recent publication in Science Advances 2020;6(49). Developed in yeast, such is the importance of identifying lipid modification in all species that I am confident this protocol will be used widely in the future.

We thank the reviewer for the appreciation of our manuscript. 

Reviewer #3:

The authors provide an excellent job of summarizing the significance and relevance of improving the protocol for detection of GPI-AP in yeast. This includes a detailed explanation of the roles of GPI and the distinctions between that of yeast and mammalian. Additionally, a comprehensive description of the rational for use of mass spectrometry-based methods versus the use of traditional methods. The authors' observations regarding the significance of using the ceramide acyl chain length for the ER protein. There are a few clarifications that I am looking for:

We thank the reviewer for the useful comments, and we hope to have answered the questions with the following.

1) How many samples were used for each the wild type and GhLag1, and just one blank were used? Were any biological or technical replicated used?

Response: To obtain the results published in Rodriguez-Gallardo et al. (2020) (Sci Adv 6(50):eaba8237) we performed five experiments, each of them with its blank, Wild-type and Ghlag1 sample. Thus, we obtained a total of 5 biological replicates by sample. For each of them each transition of the MRM list was read three times (three technical replicates). We have added this information to the manuscript.

2) Which instrument/columns were used for MS/MS?

The mass spectrometer used in our study is a TSQ Vantage and samples were infused on it using a Triversa Nanomate. We have added this information to the manuscript.

---

## [Editor Report · Decision Letter 1]

2 Aug 2021

Determination of the lipid composition of the GPI anchor

PONE-D-21-17892R1

Dear Dr. Muñiz,

We’re pleased to inform you that your manuscript has been judged scientifically suitable for publication and will be formally accepted for publication once it meets all outstanding technical requirements.

Kind regards,

Michael Polymenis, Ph.D.

Academic Editor

PLOS ONE
---

## [Editor Report · Acceptance letter]

5 Aug 2021

PONE-D-21-17892R1 

Determination of the lipid composition of the GPI anchor 

Dear Dr. Muñiz:

I'm pleased to inform you that your manuscript has been deemed suitable for publication in PLOS ONE. Congratulations! Your manuscript is now with our production department. 

Kind regards, 

on behalf of

Dr. Michael Polymenis 

Academic Editor

PLOS ONE